# Key theories and technologies and implementation mechanism of parallel computing for ternary optical computer

Sulan Zhang[1,2]*, Junwei Chen[1], Zihao Liu[1], Xiaolin Wang[1,3,4], Chunhua Zhang[1], Jun Yang[1]

1 School of Information Science and Engineering, Jiaxing University, Jiaxing, Zhejiang, China, 2 Key Laboratory of Medical Electronic and Digital Health of Zhejiang Province, Jiaxing University, Jiaxing, Zhejiang, China, 3 Shanghai Business School, University of Shanghai for Science and Technology, Shanghai, China, 4 Jujiang Construction Group Co., Ltd., Jiaxing, Zhejiang, China

* zhangsl000111@sina.com

**Data Availability Statement:** All relevant data are within the paper.

**Funding:** YESthis work is supported by National Natural Science Foundation of China under Grant

## Abstract

Ternary Optical Computer (TOC) is more advanced than traditional computer systems in parallel computing, which is characterized by huge amounts of repeated computations. However, the application of the TOC is still limited because of lack of key theories and technologies. In order to make the TOC applicable and advantageous, this paper systematically elaborates the key theories and technologies of parallel computing for the TOC through a programming platform, including reconfigurability and groupable usability of optical processor bits, parallel carry-free optical adder and the TOC's application characteristics, communication file to express user's needs and data organization method of the TOC. Finally, experiments are carried out to show the effectiveness of the present theories and technologies for parallel computing, as well as the feasibility of the implementation method of the programming platform. For a special instance, it is shown that the clock cycle on the TOC is only 0.26% of on a traditional computer, and the computing resource spent on the TOC is 25% of that on a traditional computer. Based on the study of the TOC in this paper, more complex parallel computing can be realized in the future.

## Introduction

With the continuous development of science and technology, people need faster computers. It means increasing power to chip integration levels. Since the hardware of electronic computers is limited by Moore's Law, new type computers are paid increasingly attention to, such as optical computers, quantum computers, DNA computers, etc. [1–4]. Due to the high parallelism and high information capability of light, optical computing has gradually become a research hotspot.

Optical computing has some advantages, such as numerous processor bits, high parallelism, easy expansion, low power consumption and so on [5–8]. It can solve the bottleneck problem of electronic computing. In order to take advantages of optical computing, the TOC was proposed in 2000 [1, 2]. For this optical processor, it has a mess of processor bits and these

No. 51875333 for CHEN Junwei and Natural Science Foundation of Zhejiang Province under grant No. LQ22F020004 for WANG Xiaolin and Natural Science Foundation of China under Grant No. 61672006 for none. The funders had no role in study design, data collection and analysis, decision to publish, or preparation of the manuscript.

**Competing interests:** The authors have declared that no competing interests exist.

processor bits can be bitwise reconstructed and grouped, and computational function of each processor bit can be reconfigured at runtime. Therefore, the TOC is more advantageous than traditional computer systems in solving problems that require more resources and parallel computing with large amounts of data.

After many years of exploration, researchers have made much progress on the TOC, such as the Decrease-Radix Design theory [9], the programming model [10], the MSD adder [11–14], parallel radix-4 MSD iterative division [15], Traffic Flow Model [16], MSD iterative division algorithm and implementation [17], implementation of DFT algorithm and so on [18].

Although these works are valuable and of great contribution, the advantages of optical computing have not been completely exploited, especially in the key corresponding theories and technologies of parallel computing. In order to solve this problem and make full use of the TOC's advantages, key theories and technologies of the TOC in terms of parallel computing are developed in detail in this paper, and implementation method of the parallel-computing programming platform is established. In addition, the architecture of the present study can be generalized to more complex parallel computing of the TOC.

## Materials and methods

### Related work

Many countries have started the research of optical computing from different perspectives. In medicine, optical technology is proposed for three-dimensional reconstruction of soft tissue surface geometry in minimally invasive surgery (MIS) [19, 20]. In computer science, in order to eliminate the secondary cache and increase the processor space, an all-optical memory designed with high-bandwidth optical characteristics is proposed. In chip development technology, a new type of optical chip is proposed and the chip is more efficient, faster, and consumes less power than electronic chips when performing computing tasks [21]. In high-performance computing systems, in order to meet the need for bandwidth expansion, optical interconnection technology is used. Obviously, optical computing has more application directions and application prospects.

Over years, some research scholars have tried to build some optical computer systems [1, 2, 22–26]. In Japan, the research of optical array logic appeared [27, 28], Optical computer mouse was developed [29]. In the United States, the study of dual-mode optical computers appeared [30]. In China, ternary optical computer appeared [1, 2]. In Canada, light quantum gates were designed [31]. In Germany, the technology of creating feasible reflectance data appeared [32]. In Russia, the technology of optical scanning appeared [33].

From the above description, it can be seen that many people have devoted a lot of effort to the study of optical computers from different perspectives in different countries. The paper mainly elaborates key theories and core technologies of TOC processing parallel computing.

### Key theories and technologies of parallel computing for the TOC

**The reconfigurability of optical processor bits.** The reconfigurability of optical processor bits supports parallel computing. After the Decrease-Radix Design theory was applied into the TOC, it is known that the TOC has 18 simple basic units (i.e. SBU) and no more than six of them can constitute any of the 19683 tri-valued logic operators [9]. It is also found that every SBU has the same optical structure, as shown in Fig 1. In this structure, there are two optical paths. The input data a enters the main optical path which consists of two polarizers (P1 and P2) and a liquid crystal (LC) to form a sandwich structure. The other input signal b enters the control optical path [10]. f1 and f2 are semi-reflectors, F is a holophote, V is a vertical polaroid, H is a horizontal polaroid, g1, g2 and g3 are phototubes, S is a tri-pick device, k1,

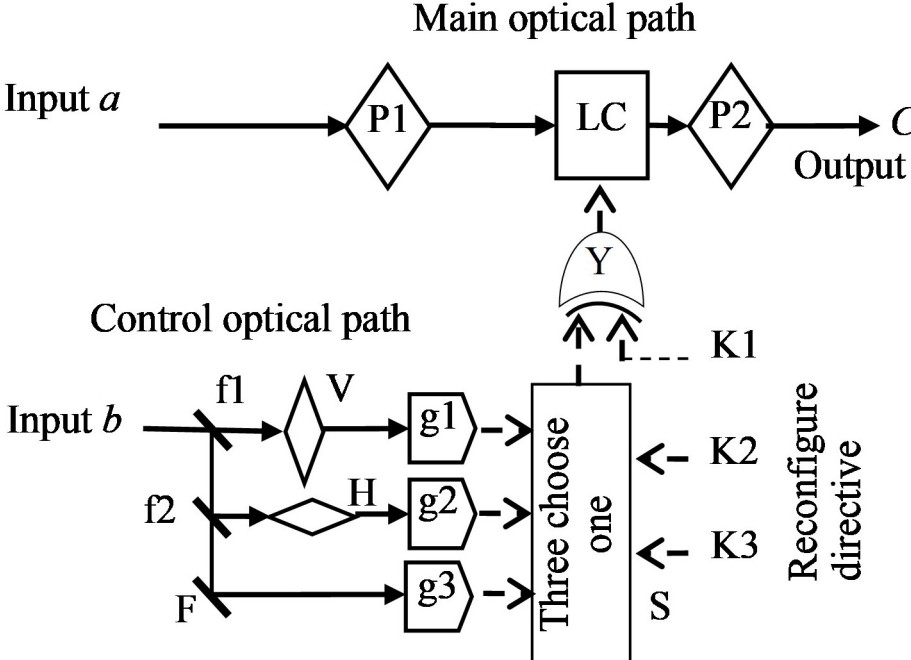

**Fig 1. Simple basic unit structure.**

k2 and k3 are reconstruction instruction bits, Y is an XOR gate. According to the values of the reconstructed instruction bits k2 and k3, S selects one of the output signals from g1, g2 and g3 and sends the selected signal to Y. Y's output signal is used to control the optical rotation of LC in the main optical path. When the reconstruction instruction bit k1 is 1, Y makes the S output signal negate, and when k1 is 0, S output signal is unchanged. The difference between SBUs is that the static rotation of LC is opposite, or polarization direction of two polarizers is different, or the characteristic of the control optical path is different [34].

According to the polarization direction of two polarizers, which is equally divided into four areas, named VV, VH, HH and HV. Based on the polarization of the polarizer, the control light path is divided into two areas (H, V). According to Decrease-Radix Design theory, when constructing a specific logic operator, first we should select the pixels of each type of main optical path, then give the values of the reconstruction instruction bits k1, k2 and k3 for these pixels [35]. After the computational task is completed, the logic operator will release all the tiny SBUs that are occupied, in order to reuse them next time. If the SBUs are enough, a Ternary Optical Processor (TOP) can be constituted. The reconfigurability of optical processor bits is the basic of parallel computing.

**Groupable usability of optical processor bits.** The carry operations are implemented bit by bit in a traditional adder, which makes the adjacent bit operation coupled. Besides, the processor bits are not grouped for different operations simultaneously, and practically utilized as a whole for a single operation.

In the TOC, a logical operation is limited to its own data bits. There is no correlation between the optical processor bits of a three-valued logical operation.

In 1950, the redundant counting method was used to eliminate the carry process in the addition [36, 37], three symbols represented redundantly with two binary digits form the Modified Signed Digit (MSD) number system [38]. Mathematicians have proved that the addition

operation of two MSD numbers can be done by a series of appropriate three-valued transformations, there is no-carry operation in these transformation process. All bit transformations are carried out in parallel [38].

The MSD adder can express the carry value in the redundant part of the bit, therefore there are not carry operations in the MSD adder [11]. Considering that the MSD addition is achieved by a set of three-valued logic transformations, MSD adder is used in TOC. So processor bits of the TOP can be used in groups. The groupable usability of optical processor bits is another key basic of parallel computing, and it enables different regions of the TOP to be configured as different operators. Optical processor bits can be selectively grouped or combined.

**Parallel carry-free optical adder.** In 2010, the problem of parallel carry-free MSD addition was considered. In 2013, a set of simpler parallel carry-free three-step MSD addition transformation rules was obtained according to the three-step TW-MSD addition transformation T, W, T', W', T2(S) [11], as shown in Table 1. In 2016, the characteristics of the parallel carry-free three-step MSD addition was described [16].

The MSD adder is a no-carry full adder in parallel. Considering that the TOP has three information symbols, each bit of the TOP can be reconstructed into any three-valued logic operator, and the number of TOP bits is large. The above features meet the requirements of MSD adder, so the TOC in the present study finishes the binary addition operation by using the MSD adder in parallel. The transformation truth table of the MSD adder is listed in Table 1. The three logical steps are independent, which are described as follows [11].

Step1: Apply operation T and W to the operands a and b bit by bit and append one 0 to the tail of the result of T.

Step2: Apply operation T' and W' separately to the results of T and result of W bit by bit. And append one 0 to the tail of the result of T'.

Step3: Apply operation S which is the same to operation T to the results of T' and result of W' bit by bit. The result is the sum of a and b.

Parallel carry-free optical adders make it realizable for the TOC to perform parallel computing. The calculation process of Parallel carry-free optical adders is divided into three steps as shown in Fig 2.

**The application characteristics of the TOC.** There are thousands of processor bits in the TOC. The computing function of each processor bit can be reconfigured according to the needs of users, and each processor bit can be assigned to different tasks. These characteristics make its hardware fully meet the requirements of the structure data computer. The simple structure data computer summarizes the application characteristics of the TOC [39]. Each

**Table 1. MSD adder's transformation truth table.**

| a | b | T | W | T' | W' | S |
|---|---|---|---|---|----|---|
| -1 | -1 | -1 | 0 | -1 | 0 | 1 |
| -1 | 0 | -1 | 1 | 0 | -1 | 0 |
| -1 | 1 | 0 | 0 | 0 | 0 | -1 |
| 0 | -1 | -1 | 1 | 0 | -1 | 0 |
| 0 | 0 | 0 | 0 | 0 | 0 | 0 |
| 0 | 1 | 1 | -1 | 0 | 1 | -1 |
| 1 | -1 | 0 | 0 | 0 | 0 | -1 |
| 1 | 0 | 1 | -1 | 0 | 1 | 0 |
| 1 | 1 | 1 | 0 | 1 | 0 | 1 |

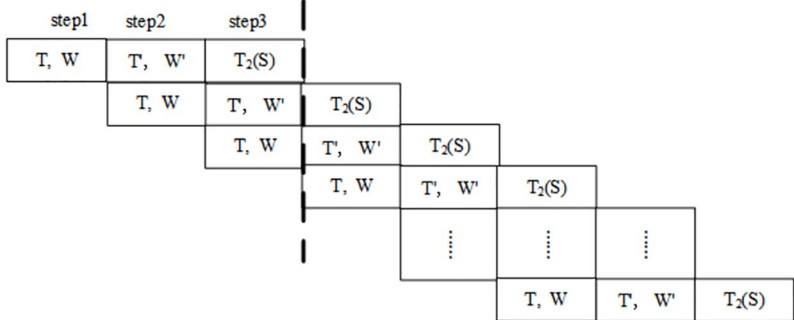

**Fig 2. The calculation process of parallel carry-free optical adders.**

expansion of data types pushes computers into a new field of application. Especially for the processing of structured data, computers have been brought into a broad field of computing and management. Structured data computer can directly process structured data. The structured data composed of different simple data types can be executed in one operation instruction. At present, the ternary optical processor has ability to operate the structured data independently. For example, we define a structure data type called JXDX:

```
JXDX {
        float a;
        int b;
        char c;
        float d[8];
}
```

In the curly braces, each line is one component of the structured data type JXDX. The fourth line, float d [8], is an array of float data type and the rest are simple data types, including float type, int type and char type. All those components are the simple data types of the C programming language. The TOC can operate the structure data type called JXDX in different optical processor bits.

Before the calculation, the TOC processor bits need to be reconstructed according to the user's calculation requirements, so we include the user's calculation data and operator types in a communication file, and the TOC management software obtains the corresponding parameters by parsing the communication file. The concept of simple structure data computer makes the TOC a continuation and extension of electronic computer in terms of computing power.

**Communication file.**　To solve different problems, there are various files formats. The PDF file is to solve the file exchange problem in the form of picture, the DOC file is to solve the text editing problem, the ELF file is to solve the linking problem of multiple executable files, the communication file (SZG file) is to solve the problem of processor bits allocation and computing function reconstruction in the TOP. These files contain the same file structure: the file header and the file body. Various files have some structured information, such as file name, file length, file password, and some inconveniently structured contents. In order to accommodate the computer system to identify and manage these files, structured information is traditionally stored at the beginning of the file, called file header, and the inconvenient expression contents are stored at the end of the file, called file body. We also follow the convention. The key difference between different files is that the definition of each field in file header

is different. The key technology is whether the definition of each field in file header can make its function complete.

Currently, communication file is the only way to express the user's needs about calculation rules and data bits in the TOC's application [40]. Programmers must follow a special format to create a communication file to the TOC's system software find user's demands from the file. So the communication file format is the key of writing the TOC's application program.

The new simplified version of communication file format has two parts: indication section and data section as shown in Fig 3. The indication section includes user's information and demands for the TOP. The data section includes all of original data in *.SZG file or all of computing results in *-R.SZG file.

In Fig 3, in the indication section, there are one File mark and n+1 Labels (Calculation labels) ranging from 0 to n. The File mark records the version of the communication file, the communication filename, user's address and Label amount in the communication file. Each Label records a user's demand, which contains the calculation rule, data bit's number, data amount and the first data address in the data section. Obviously, programmers can at the most give n+1 calculation needs in a communication file. In the *.SZG, the original data of each Label are stored in the data section one by one, and the results are stored in the *-R.SZG. The starting address of each original data or result region is recorded in the First data address of corresponding Label.

**Data organization method of the TOC.**   The user's calculation data and requirements are included in the communication file. The user program transmits this file to the TOC, and the TOC task management software parses the file according to the format of the communication file to obtain the corresponding calculation data and calculation rule. And it assigns the corresponding processor bits based on the number of data bits and calculation rule. In order to improve the calculation efficiency of the TOP, the TOC task management software organizes the corresponding calculation data according to the number allocated by the calculator. For example, if there are 2 operations, namely $p_1 = x + y$, $p_2 = z \vee q$, here, $x$, $y$, $z$ and q all are int. and the amount of data is 10000 and 20000 respectively, then the p1 operation is assigned 1 adder, and the p2 operation is assigned 2 logical OR operators, then a pair of data frames of calculated data should be organized into the following format in Fig 4.

Data A and Data B are equivalent to two structured data as shown in Fig 5.

When calculating, it is only necessary to sequentially send the organized data into the TOP.

## Implementations method of parallel-computing programming platform based on the TOC

In this section, a three-step MSD adder is used in the TOC, the calculation is accomplished in pipeline as shown in Fig 2. In optical implementation, horizontal polarized light (H), the vertical polarized light (V) and no light (W) are used. H represents u (-1). V represents 1. W represents 0.

In order to better understand these theories and technologies, the specific steps for implementation of user's parallel calculations are as follows:

Step 1: Organize all user's calculation rules and original data into a communication file.
According to a specific instance, user inputs relevant information through the communication file input interface [40], then generates a communication file.

Step 2: The communication file is sent to the TOC.

Run the user's program and send the generated communication file to the TOC. Step 3: The optical processor bits are allocated in groups.

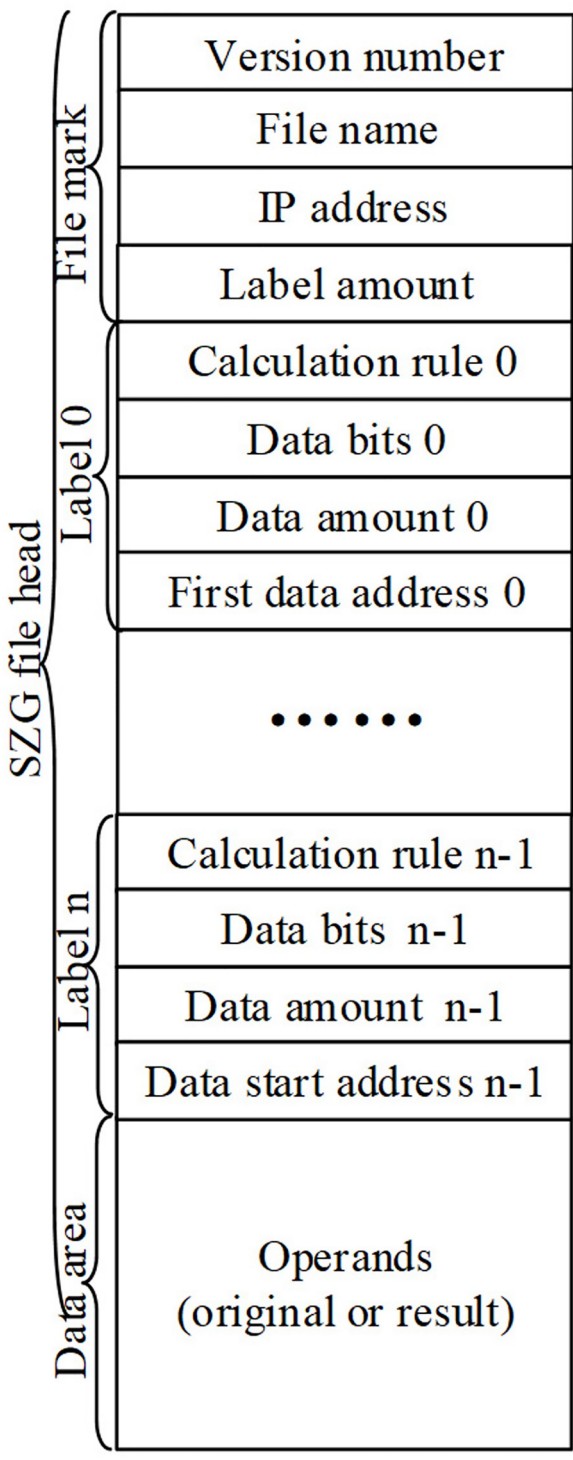

**Fig 3. Communication file format.**

| Data A | Data B |
|--------|--------|
| $x_0$ | $y_0$ |
| $z_0$ | $q_0$ |
| $z_1$ | $q_1$ |
| $x_1$ | $y_1$ |
| $z_2$ | $q_2$ |
| $z_3$ | $q_3$ |
| ...... | ...... |
| ...... | ...... |
| ...... | ...... |
| $x_{9999}$ | $y_{9999}$ |
| $z_{19998}$ | $q_{19998}$ |
| $z_{19999}$ | $q_{19999}$ |

**Fig 4. Data organization format.**

The TOC task management software receives the communication file and parses it, the optical processor bits are allocated in groups according to the calculation rules required by the user in the communication file.

Step 4: The TOP is reconstructed to form a structured computer required by users. When the TOP reconstruction command is started, these allocated optical processor bits are reconstructed into required operators by users according to calculation rules in the communication file.

Step 5: Generate structured data for the TOP.

```
A {                                  B {
    int   x_i;                           int   x_i;
    int   z_i;                           int   z_i;
    int   z_{i+1};                       int   z_{i+1};
}                                    }
```

**Fig 5. Structured data A and B.**

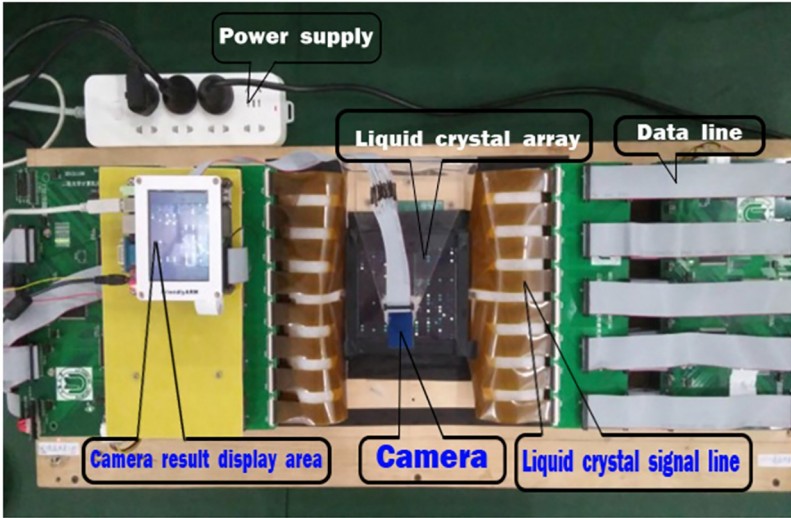

**Fig 6. SD16-the TOC prototype system.**

The TOC task management software organizes the original data in the communication file into structured data according to the order of required operators.

Step 6: Send these organized structured data to these reconstructed operators in the TOP. These organized structure data are sent into these reconstructed operators to calculate. Then the optical results are obtained.

Step 7: The optical results are sent to the decoder by the optical processor control software, and the decoder reconvert the optical result data into electrical signal data in binary.

Step 8: Repeat step 5 to step 7 until all original data in communication file has been calculated.

Finally, these electrical signal result data are collected into *–R.SZG file and then returned to the user's program.

## Results and discussion

### Experiments and analysis of the TOC handling a parallel computing

**Experiments environment.** The experiment system is the TOC-SD16, the architecture of which is shown in Fig 6. It includes two parts: the master computer and the slave computer. The master computer is an EC with a 64bit windows 8.1 computing system, and Pentium(R) Dual-Core CPU @3.00GHz, 4GB DDR4 memory. The TOC monitoring-system runs on the master computer, the main task of which is to receive the user's original communication file, adjust and encode the user's input data, and then generate the operation control communication file. The slave computer is to control and process operations based on its original communication file and middle operation control file. It includes three function modules, the encoder, optical processor (the arithmetic unit), and the decoder. The encoder consists of an LCD and a vertical polarizer. The decoder consists of five components: a vertical polarizer, a horizontal polarizer, spectroscope, and two photoelectric converters. The optical processor consists of two components: two pieces of polarizer and one LCD with 576 pixels which means most 576 bits' data can be processed in parallel. ARM9 is used to encode the results and handle

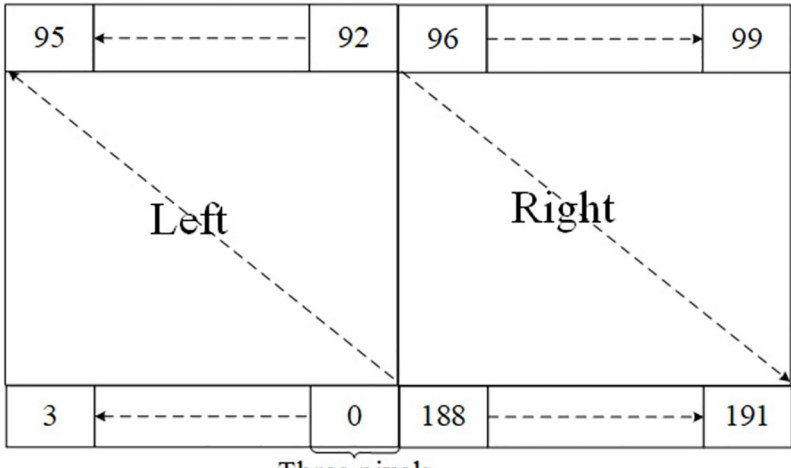

**Fig 7. Liquid crystal divisions.**

the data for feedback. The control light path is divided into two areas (H, V), and the main light path is divided into four areas (HH, HV, VH, VV).

The structure of the optical processor of the TOC is shown in Fig 7. It consists of two parts, the left and right part, and each pixel of LCD in each part is indicated as the arrow. It has 576 pixels, in SD16, in order to easily observe the result data. The three adjacent pixels in the same row form a processor bit. That is, there are 192 processor bits.

**Experiment process and analysis.** Four kinds of operations are tested, i.e., $f_1 = a + b, f_2 = c - d, f_3 = e \wedge g$ and $f_4 = h \vee i$. The four operations and corresponding data are contained in ZH.SZG. For f1, there are a total of 200000 pairs of original data, where a and b are 8-digit MSD number; for f2, there are 200000 pairs of data, where c and d are 8-digit MSD number; for f3, there are 100000 pairs of data, where e and g are 8-digit ternary number; and for f3, there are 100000 pairs of data, where h and i are 8-digit ternary number.

For step 1 and step 2, please refer to Ref. [10].

Step 3: Processor bits for ZH.SZG are allocated, and these processor bits are reconstructed into the user's composite operator.

The TOC task management software receives ZH.SZG and parses it, then +, -, $\wedge$ and $\vee$ four calculation rules and the corresponding data number, 200000, 200000, 100000, 100000 are obtained. Here a and b are 8-digit tri-value numbers, requiring 37 TOP bits to construct a 8-digit TW-MSD parallel adder to accommodate the "+" operator $f_1$; c and d are 8-digit tri-value numbers, requiring 37 TOP bits to construct a 8-digit TW-MSD parallel adder accommodate the "-" operator $f_2$; e and g are 8-digit tri-value logic data, requiring 8 TOP bits to construct the 8-digit $\wedge$ operator $f_3$; similarly, 8 TOP bits are required to construct the 8- digit $\vee$ operator $f_4$[17].

In the present case, the data amounts of f1 and f2 are defined twice those of f3 and f3. Therefore, if the number of the operators in terms of executing f1 and f2 in the TOP is twice that of f3 and f4, the consumed time of the former operation will be close to the latter. So the data bit allocation module will assign 88 (= 44 * 2) processor bits to f1, constructing two f1 "+" operators; 88 (= 44 * 2) processor bits are assigned to f2 for constructing two f2 "-" operators; similarly, 8 processor bits are assigned to f3; then 8 processor bits are assigned to f3, so which forms a composite operator consisting of 192 (= 88 + 88 + 8 + 8) optical processor bits.

**Table 2. Processor bit allocation of the example.**

| T | W | T' | W' | S | AND | OR |
|---|---|---|---|---|---|---|
| 0-7($f_2$) | 32-39($f_2$) | 64-72($f_2$) | 100-108($f_2$) | 136-145($f_2$) | 176-183($f_3$) | 184-191($f_3$) |
| 8-15 ($f_2$) | 40-47 ($f_2$) | 73-81 ($F_2$) | 109-117 ($f_2$) | 146-155 ($f_2$) | | |
| 16-23($F_1$) | 48-55 ($F_1$) | 82-90 ($F_1$) | 118-126 ($f_1$) | 156-165 ($f_1$) | | |
| 24-31($F_1$) | 56-63 ($F_1$) | 91-99 ($F_1$) | 127-135 ($f_1$) | 166-175 ($f_1$) | | |

Considering the pipelined computing technology, one result is gotten in every clock cycle, except the first two clocks, this composite operator only need to run 100002 (=100000+2) times to complete all the data calculation. The pipeline implementation of TW-MSD adder is used. Data bit allocation of the example is shown in Table 2.

Step 4: Optical operators required by the user is reconstructed. In other words, the structured computer required by the user is formed.

Each pixel may output no-light (W), horizontal polarized light (H) or vertical polarization light (V), the light intensity of W, H and V may be different. For a processor bit, there is only one output (W or V or H). Each operation corresponds to a refactoring instruction, for example, the refactoring instruction of T is 011 101 110 000 001 010. Refactoring instructions of 192 processor bits form a reconstructed latent image table, which is sent into reconstructor, the TOP completes reconstitution. These optical operators required by the users are referred to as compound operators, it is equivalent to a structured computer.

Step 5: Structured data of the composite operator are generated.

The original data in ZH.SZG are fetched and these data are organized into structured data. Here, the structured data of the first 3 screens are given. u represents -1.

For the first screen, one structured data a in decimal is 57, 39, 181, -154. The corresponding MSD number is 00111001, 00100111, 110u0101, u0u010u0 respectively. Another structured data b in decimal is 35, 84, -9, -46. The corresponding MSD number is 00100011, 01010100, 0u110111, u10101u0 respectively.

For the second screen, one structured data a in decimal is 101, 72, -37, 161. The corresponding MSD number is 01100101, 01001000, u11u1011, 1010001u respectively. Another of structured data b in decimal is 120, 163, 245, 195. The corresponding MSD number is 01111000, 10100011, 11110101, 110001u1 respectively.

For the third screen, logic calculation is added. One of structured data a in decimal is 92, 215, -85, -120,180, -26. The corresponding MSD number is 01011100, 11010111, u011u011, u01uu000, 110u0100, 00u010u0 respectively. Another structured data b in decimal is 40, 199, 110, 102, -10, -82. The corresponding MSD number is 00101000, 11000111, 10u01110, 011010u0, 0u110110, 010101u0 respectively.

Step 6: These organized structured data are sent to corresponding reconstructed operators in the TOP. Then the resultant optical data are obtained.

The computing process of the first three screens is captured as and shown in Fig 8. In Fig 8 (a)–8(c) are respectively the experiment results of the first, second and third screen. The TOC and theoretical results are compared in Tables 3–5, respectively. They match perfectly with each other C is the result of the operand a and the operand b.

Step 7: The optical results are sent to the decoder, and the decoder reconvert the optical result data into electrical signal data in binary.

Step 8: Repeat step 5 to step 7 until all original data in communication file has been calculated.

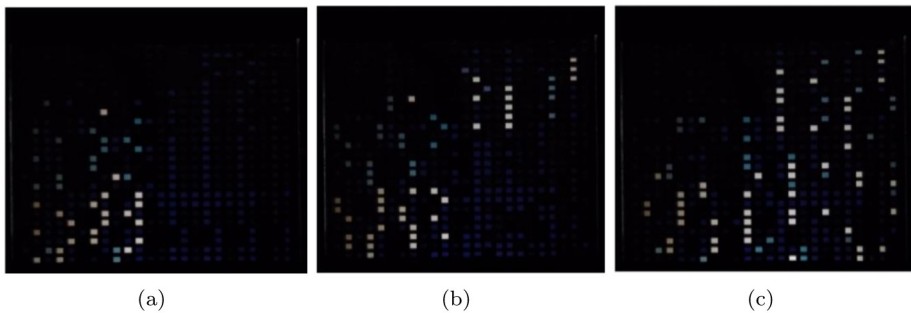

(a)                                        (b)                                        (c)

**Fig 8. The experiment result of the first three screen.**

Finally, these electrical signal result data are collected into ZH–R.SZG file, and it is then returned to user's program.

The experiment results on every screen are correct, which validates the theories and technologies of parallel computing.

## Discussion

A basic module of the optical processor of SD16 has 192 processor bits, and it can be allocated again to another basic module to construct another identical composite operator, so it only needs to run 50000 times, before all the operations are completed. The optical processor of the TOC can be installed up to 64 basic modules, which can be constructed 64 identical composite operators, it only needs 100000/64 = 1563 times operations to complete all the calculation task. And for electronic processor, it needs to perform 600000 times operations for the task.

**Table 3. Comparison between experiment and theory results for the first screen.**

| C | Original data 1 | | $T_1$ | | $W_1$ | | $T_1'$ | | $W_1'$ | | $S(T2)$ | |
|---|---|---|---|---|---|---|---|---|---|---|---|---|
| | a | b | TOC | theory | TOC | theory | TOC | theory | TOC | theory | TOC | theory |
| $f_1$ | 00111001 | 00100011 | 00111011 | 00111011 | 000uu0u0 | 000uu0u0 | - | - | - | - | - | - |
| $f_1$ | 00100111 | 01010100 | 01110111 | 01110111 | 0uuu00uu | 0uuu00uu | - | - | - | - | - | - |
| $f_2$ | 110u0101 | 0u110111 | 10100111 | 10100111 | u0u000u0 | u0u000u0 | - | - | - | - | - | - |
| $f_2$ | u0u010u0 | u10101u0 | u1u111u0 | u1u111u0 | 0u1uuu00 | 0u1uuu00 | - | - | - | - | - | - |
| $f_3$ | - | - | - | - | - | - | - | - | - | - | - | - |
| $f_4$ | - | - | - | - | - | - | - | - | - | - | - | - |

**Table 4. Comparison between experiment and theory results for the second screen.**

| C | Original data 2 | | $T_2$ | | $W_2$ | | $T_2'$ | | $W_2'$ | | $S(T2)$ | |
|---|---|---|---|---|---|---|---|---|---|---|---|---|
| | a | b | TOC | theory | TOC | theory | TOC | theory | TOC | theory | TOC | theory |
| $f_1$ | 00111001 | 00100011 | 00111011 | 01111101 | 000uuu0u | 000uuu0u | 000000000 | 000000000 | 00110u100 | 00110u100 | - | - |
| $f_1$ | 01001000 | 10100011 | 11101011 | 11101011 | uuu0u0uu | uuu0u0uu | 000000000 | 000000000 | 0100u110u | 0100u110u | - | - |
| $f_2$ | u11u1011 | 11110101 | 01101111 | 01101111 | 0000uuu0 | 0000uuu0 | 000000000 | 000000000 | 1u1u01100 | 1u1u01100 | - | - |
| $f_2$ | 1010001u | 110001u1 | 11100100 | 11100100 | 0uu00u00 | 0uu00u00 | 00u100u00 | 00u100u00 | u10000000 | u10000000 | - | - |
| $f_3$ | - | - | - | - | - | - | - | - | - | - | - | - |
| $f_4$ | - | - | - | - | - | - | - | - | - | - | - | - |

**Table 5. Comparison between experiment and theory results for the third screen.**

| C | Original data 3 | | $T_3$ | | $W_3$ | | $T'_3$ | | $W'_3$ | | $S(T2)$ | |
|---|---|---|---|---|---|---|---|---|---|---|---|---|
| | a | b | TOC | theory | TOC | theory | TOC | theory | TOC | theory | TOC | theory |
| $f_1$ | 01011100 | 00101000 | 01111100 | 01111100 | 0uuu0u00 | 0uuu0u00 | 000000000 | 000000000 | 011100u1u | 011100u1u | 000110u100 | 000110u100 |
| $f_1$ | 11010111 | 11000111 | 11010111 | 11010111 | 000u0000 | 000u0000 | 000000000 | 000000000 | 100u1u10u | 100u1u10u | 00100u110u | 00100u110u |
| $f_2$ | u011u011 | 10u01110 | 00010111 | 00010111 | 000u0u0u | 000u0u0u | 000000000 | 000000000 | 011010000 | 011010000 | 01u1u01100 | 01u1u01100 |
| $f_2$ | u01uu000 | 011010u0 | u11u00u0 | u11u00u0 | 1u010010 | 1u010010 | 000000000 | 000000000 | 110u01u00 | 110u01u00 | 0u0100u000 | 0u0100u000 |
| $f_3$ | 110u0100 | 0u110110 | - | - | - | - | - | - | - | - | 0u0u0100 | 0u0u0100 |
| $f_4$ | 00u010u0 | 010101u0 | - | - | - | - | - | - | - | - | 01u111u0 | 01u111u0 |

**Table 6. Performance comparison between electronic processor and the TOP of the example.**

| Performance Index | Electronic Processor | | Composite operator | |
|---|---|---|---|---|
| | 32-bit | 64-bit | A | 64 |
| Reconsturciton time ($T_0$) | 0 | 0 | 1 cycle | 1 cycle |
| Processor bits ($W$) | 4 * 32 = 128 | 4 * 64 = 256 | 192 | 192 * 64 = 12288 |
| Operation (Clock) cycles($T_y$) | 600000 | 600000 | 100002 | $\lceil 100000/64 \rceil$ + 2 = 1565 |
| Amount of resource ($W * T_y + T_0$) | 76800000 | 153600000 | 19200384 + $T_0$ | 19230720 + $T_0$ |

For example, a three-step MSD adder is used in the TOC, the calculation is accomplished in pipeline. The reconstruction time of the TOP is 1 clock cycle, it takes 3 clock cycles to complete an addition operation. An operation cycle is assumed to be a clock cycle. The computational efficiency of the TOP and electronic processor can be compared in resource utilization and consumed time, as shown in Table 6. It is shown that the clock cycle on the TOC is only 0.26% of on a traditional computer, and the computing resource spent on the TOC is 25% of that on a traditional computer.

The TOP needs to be reconstructed. If the calculation data is small, it is not suitable to use the TOC for calculation. The original calculation data are more, the efficiency of the TOP is higher. Therefore, the TOC is suitable for calculations with large amounts of data.

## Conclusion

The TOC is a structured data computer. That is, the TOC can process multiple kinds of data in parallel. For this purpose, the authors develop key theories and technologies including reconfigurability and group usability of optical processor bits, parallel carry-free optical adder and the application characteristics of the TOC, communication file and data organizing method of the TOC in parallel computing. For the first time, data organizing format of the TOC is designed and implementation method of parallel-computing programming platform on the TOC is carried out in the present paper. Finally, the experiments of TOC handling a parallel computing are implemented, which verifies the proposed theories and technologies. It is shown that the computing efficiency of the TOP is greatly improved in resource utilization and consumed time, compared with a traditional computer where the processor bits are unable to be grouped and allotted for different users. Thus the advantages of the TOC can be further generalized to complex parallel computing.

## Acknowledgments

Thank team members of our research group for their kind help and valuable discussions in preparing the paper. We also thank the Research Center of the TOC at Shanghai University for the valuable equipment.

Sulan ZHANG wrote the original manuscript. Junwei CHEN and Jun YANG edited the manuscript. Zihao LIU, Xiaolin WANG and Chunhua ZHANG check the manuscript.

## Author Contributions

**Data curation:** Zihao Liu.

**Software:** Sulan Zhang, Xiaolin Wang.

**Writing – original draft:** Sulan Zhang.

**Writing – review & editing:** Junwei Chen, Chunhua Zhang, Jun Yang.

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
