## [Decision Letter · Decision Letter 0]

14 Feb 2023

PONE-D-22-31631Key Theories and Technologies and Implementation Mechanism of Parallel Computing for Ternary Optical ComputerPLOS ONE

Dear Dr. ZHANG,

Thank you for submitting your manuscript to PLOS ONE. After careful consideration, we feel that it has merit but does not fully meet PLOS ONE’s publication criteria as it currently stands. Therefore, we invite you to submit a revised version of the manuscript that addresses the points raised during the review process.

We look forward to receiving your revised manuscript.

Kind regards,

Talib Al-Ameri, Ph.D

Academic Editor

PLOS ONE

Journal Requirements:

YES，this work is supported by National Natural Science Foundation of China under Grant No. 51875333 for CHEN Junwei and Natural Science Foundation of Zhejiang Province under grant No. LQ22F020004 for WANG Xiaolin and Natural Science Foundation of China under Grant No. 61672006 for none.

Reviewers' comments:

Reviewer's Responses to Questions

**Comments to the Author**

1. Is the manuscript technically sound, and do the data support the conclusions?

Reviewer #1: Yes

Reviewer #2: Yes

2. Has the statistical analysis been performed appropriately and rigorously? 

Reviewer #1: Yes

Reviewer #2: Yes

3. Have the authors made all data underlying the findings in their manuscript fully available?

Reviewer #1: Yes

Reviewer #2: Yes

4. Is the manuscript presented in an intelligible fashion and written in standard English?

Reviewer #1: Yes

Reviewer #2: No

5. Review Comments to the Author

Reviewer #1: The paper introduced the related technologies to exert the ability of TOC parallel computing, and gave an experimental comparison. The scientific content and the results presented in the paper are adequate for publication.

Reviewer #2: Ternary Optical Computer (TOC) is more advanced than traditional computer systems in parallel computing. Based on huge amounts of repeated computations, this paper systematically elaborates the key theories and technologies of parallel computing for the TOC through a programming platform. And experiments are carried out to show the effectiveness of the present theories and technologies for parallel computing, as well as the feasibility of the implementation method of the programming platform. Based on the study of the TOC in this paper, more complex parallel computing can be realized in the future. The scientific content and the results presented in the paper are interesting and valuable. However, I have some suggestions as follows.

(1) In the second paragraph in the section 3.1, please give the full name of TOP.

(2) In section 3.3, the working process of parallel carry-free optical adder should be charted to show the details.

(3) Pay attention to writing details, for example: the second Paragraph in the section 3.3, in the sentence “Considering that TOP has three ...” , TOP should be the TOP.

The paper can be accepted for publication after the authors finish the above modifications.

6. PLOS authors have the option to publish the peer review history of their article (what does this mean?). If published, this will include your full peer review and any attached files.

Reviewer #1: No

Reviewer #2: No

---

## [Author Response · Author response to Decision Letter 0]

9 Mar 2023

List of Changes in Responses to Editors’ and Reviewers’ Comments on “Key Theories and Technologies and Implementation Mechanism of Parallel Computing for Ternary Optical Computer”

Dear Ph.D. Talib Al-Ameri,

On behalf of my co-authors, we thank you very much for all you have done on the manuscript entitled “Key Theories and Technologies and Implementation Mechanism of Parallel Computing for Ternary Optical Computer” (ID: PONE-S-22-37459). Meanwhile, we also appreciate you and reviewers very much for your helpful and constructive comments and suggestions on our manuscript. We have studied comments carefully and have made correction. The Revisions are marked in the paper. The uploaded please find the revised version. Should you have any questions please do not hesitate to let us know. We would like to express our great appreciation to you and the reviewers for the comments and suggestions on our paper.

Looking forward to hearing from you.

Best Regards,

Sulan ZHANG

Corresponding author:

Name: Sulan ZHANG 

E-mail: zhangsl000111@163.com

List of Responses

Dear Ph.D. Talib Al-Ameri and Reviewers:

Thank you very much for your letter and the reviewers’ comments and suggestions concerning on our manuscript entitled “Key Theories and Technologies and Implementation Mechanism of Parallel Computing for Ternary Optical Computer” (ID: PONE-S-22-37459). Those comments and suggestions are all valuable and very helpful for revising and improving our paper, as well as the important guiding significance to our researches. We have studied comments carefully and have made corrections which we hope meet with approval. Revised portions are marked in blue in the Revised Manuscript. The main corrections in the paper and the responds to the reviewer’s comments are as follows:

1.General revisions

There are some modifications. List of Changes in the revised paper as follows:

1.In the 26th line, TOC--- the TOC

2.In the 29th line, TOC--- the TOC

3.The 30th line was added.

4.In the 81th line, TOP--- Ternary Optical Processor (TOP)

5.In the 109th line, TOP --- the TOP

6.In the 116th line, “Step 1:” was added.

7.In the 118th line, “Step 2:” was added.

8.In the 120th line, “Step 3:” was added.

9.In the 124th line, the sentence “The calculation process of Parallel carry-free optical adders is divided into three steps as shown in Fig 2” and the Fig 2 were added.

10.In the 201-202th line, the Fig 5 was added. 

11.In the 238th line, “Results and discussion” was added.

12.In the 324th line, “C is the result of the operand a and the operand b.” was added.

More detailed modifications please refer to the Revised Manuscrip.

2.Responds to the Reviewers

Reviewer #1:

Reviewer #1: The paper introduced the related technologies to exert the ability of TOC parallel computing, and gave an experimental comparison. The scientific content and the results presented in the paper are adequate for publication.

Response to Reviewer #1: 

I would like to thank the reviewer for his affirmation of this research

Reviewer #2: 

Ternary Optical Computer (TOC) is more advanced than traditional computer systems in parallel computing. Based on huge amounts of repeated computations, this paper systematically elaborates the key theories and technologies of parallel computing for the TOC through a programming platform. And experiments are carried out to show the effectiveness of the present theories and technologies for parallel computing, as well as the feasibility of the implementation method of the programming platform. Based on the study of the TOC in this paper, more complex parallel computing can be realized in the future. The scientific content and the results presented in the paper are interesting and valuable. However, I have some suggestions as follows.

(1) In the second paragraph in the section 3.1, please give the full name of TOP.

(2) In section 3.3, the working process of parallel carry-free optical adder should be charted to show the details.

(3) Pay attention to writing details, for example: the second Paragraph in the section 3.3, in the sentence “Considering that TOP has three ...” , TOP should be the TOP.

The paper can be accepted for publication after the authors finish the above modifications.

First I would like to thank the reviewer for his affirmation of this research. The reply for the reviewer #2 is as follows:

Question 1: In the second paragraph in the section 3.1, please give the full name of TOP.

Response to Reviewer #2: 

Thanks a lot for your suggestion. In 81th line , the full name of TOP was given, it is marked in blue.

Question 2: In section 3.3, the working process of parallel carry-free optical adder should be charted to show the details.

Response to Reviewer #2: 

In 124th line , the working process of parallel carry-free optical adder should be charted to show the details as shown Fig 2.

Question 3: Pay attention to writing details, for example: the second Paragraph in the section 3.3, in the sentence “Considering that TOP has three ...” , TOP should be the TOP.

Response to Reviewer #2: 

There are some modifications. List of Changes in the Revised Manuscript as follows: 

1.In the 26th line, TOC--- the TOC

2.In the 29th line, TOC--- the TOC

3.The 30th line was added.

4.In the 81th line, TOP--- Ternary Optical Processor (TOP)

5.In the 109th line, TOP --- the TOP

6.In the 116th line, “Step 1:” was added.

7.In the 118th line, “Step 2:” was added.

8.In the 120th line, “Step 3:” was added.

9.In the 124th line, the sentence “The calculation process of Parallel carry-free optical adders is divided into three steps as shown in Fig 2” and the Fig 2 were added.

More detailed modifications please refer to the Revised Manuscrip.

---

## [Decision Letter · Decision Letter 1]

6 Apr 2023

Key Theories and Technologies and Implementation Mechanism of Parallel Computing for Ternary Optical Computer

PONE-D-22-31631R1

Dear Dr. ZHANG,

We’re pleased to inform you that your manuscript has been judged scientifically suitable for publication and will be formally accepted for publication once it meets all outstanding technical requirements.

Kind regards,

Talib Al-Ameri, Ph.D

Academic Editor

PLOS ONE

**Comments to the Author**

1. If the authors have adequately addressed your comments raised in a previous round of review and you feel that this manuscript is now acceptable for publication, you may indicate that here to bypass the “Comments to the Author” section, enter your conflict of interest statement in the “Confidential to Editor” section, and submit your "Accept" recommendation.

Reviewer #1: All comments have been addressed

Reviewer #2: All comments have been addressed

2. Is the manuscript technically sound, and do the data support the conclusions?

Reviewer #1: Yes

Reviewer #2: Yes

3. Has the statistical analysis been performed appropriately and rigorously? 

Reviewer #1: Yes

Reviewer #2: Yes

4. Have the authors made all data underlying the findings in their manuscript fully available?

Reviewer #1: Yes

Reviewer #2: Yes

5. Is the manuscript presented in an intelligible fashion and written in standard English?

Reviewer #1: Yes

Reviewer #2: No

6. Review Comments to the Author

Reviewer #1: (No Response)

Reviewer #2: (No Response)

7. PLOS authors have the option to publish the peer review history of their article (what does this mean?). If published, this will include your full peer review and any attached files.

Reviewer #1: No

Reviewer #2: No

---

## [Editor Report · Acceptance letter]

27 Apr 2023

PONE-D-22-31631R1 

Key Theories and Technologies and Implementation Mechanism of Parallel Computing for Ternary Optical Computer  

Dear Dr. ZHANG:

I'm pleased to inform you that your manuscript has been deemed suitable for publication in PLOS ONE. Congratulations! Your manuscript is now with our production department. 

Kind regards, 

on behalf of

Dr. Talib Al-Ameri 

Academic Editor

PLOS ONE